# Influence of Lipid Composition of Cationic Liposomes 2X3-DOPE on mRNA Delivery into Eukaryotic Cells

**DOI:** 10.3390/pharmaceutics15010008

**Published:** 2022-12-20

**Authors:** Vera Vysochinskaya, Sergey Shishlyannikov, Yana Zabrodskaya, Elena Shmendel, Sergey Klotchenko, Olga Dobrovolskaya, Nina Gavrilova, Darya Makarova, Marina Plotnikova, Ekaterina Elpaeva, Andrey Gorshkov, Dmitry Moshkoff, Mikhail Maslov, Andrey Vasin

**Affiliations:** 1Smorodintsev Research Institute of Influenza, 15/17 Ulitsa Professora Popova, 197376 St. Petersburg, Russia; 2Institute of Biomedical Systems and Biotechnology, Peter the Great Saint Petersburg Polytechnic University, 29 Ulitsa Polytechnicheskaya, 194064 St. Petersburg, Russia; 3Lomonosov Institute of Fine Chemical Technologies, MIREA—Russian Technological University, 86 Vernadsky Ave, 119571 Moscow, Russia; 4Global Virus Network (GVN), 725 W Lombard St, Baltimore, MD 21201, USA

**Keywords:** cationic lipids, mRNA, transfection, liposomes, non-viral delivery systems

## Abstract

The design of cationic liposomes for efficient mRNA delivery can significantly improve mRNA-based therapies. Lipoplexes based on polycationic lipid 1,26-bis(cholest-5-en-3β-yloxycarbonylamino)-7,11,16,20-tetraazahexacosane tetrahydrochloride (2X3) and helper lipid 1,2-dioleoyl-sn-glycero-3-phosphoethanolamine (DOPE) were formulated in different molar ratios (1:1, 1:2, 1:3) to efficiently deliver model mRNAs to BHK-21 and A549. The objective of this study was to examine the effect of 2X3-DOPE composition as well as lipid-to-mRNA ratio (amino-to-phosphate group ratio, N/P) on mRNA transfection. We found that lipoplex-mediated transfection efficiency depends on both liposome composition and the N/P ratio. Lipoplexes with an N/P ratio of 10/1 showed nanometric hydrodynamic size, positive ζ potential, maximum loading, and transfection efficiency. Liposomes 2X3-DOPE (1:3) provided the superior delivery of both mRNA coding firefly luciferase and mRNA-eGFP into BHK-21 cells and A549 cells, compared with commercial Lipofectamine MessengerMax.

## 1. Introduction

The potency of RNA-based therapy is determined by its delivery to target cells, along with correctly designed RNA sequences [1,2]. Nowadays, lipid nanoparticles (LNPs) are the most promising delivery vehicles of nucleic acid (NA)-based therapeutics [3,4]. LNPs are composed of four components: cationic or ionizable lipids, helper lipid, and polyethylene glycol-lipid (PEG-lipid). The first FDA-approved siRNA (small interfering RNA) drug was patisiran (Onpattro^®^), formulated in LNPs for the treatment of hereditary transthyretin-mediated (hATTR) amyloidosis, and developed by Alnylam Pharmaceuticals [5]. LNPs have also been FDA-approved for clinical use as part of COVID-19 mRNA vaccines, including Pfizer/BioNTech/Acuitas and Moderna [6]. Despite the rapid development and approval by the FDA of several LNP-based gene therapies, the optimization of LNP composition, as well as the study of physicochemical properties and biological effects, is still relevant. The influence of each LNP component and their physicochemical properties on cellular uptake and NA transfection efficiency into target cells has been demonstrated in order to optimize biological activity [7,8].

Polycationic amphiphiles based on cholesterol and natural polyamines are considered promising components of liposomal NA delivery vehicles [9,10,11,12,13]. For example, the main structural component of liposomes—1,26-bis(cholest-5-en-3β-yloxycarbonylamino)-7,11,16,20-tetraazahexacosane tetrahydrochloride (2X3)—is a combination of two structural domains, hydrophobic and hydrophilic polycationic, which are connected by spacer groups using a carbamate-type linker. Due to their positive charge, 2X3 molecules can form complexes with negatively charged NA. These NA delivery systems have shown their efficacy both in vitro and in vivo [9,10,11,12,13,14].

Helper lipids are often added to the composition of cationic liposomes to increase transfection efficiency and facilitate the escape of NA from the endosomes [15,16]. It has been shown that 1,2-dioleoyl-sn-glycero-3-phosphoethanolamine (DOPE), a zwitterionic helper lipid, is able to form an inverted hexagonal phase at the acidic pH of endosomes, which increases the efficiency of the NA molecules’ release into the cytoplasm [17,18]. It has also been shown that NA transfection efficiency depends on the molar ratio of cationic lipids to the helper lipid DOPE; in addition, such ratios vary for different types of NA (plasmid DNA, mRNA, and siRNA) [13,19,20,21,22,23].

The charge ratio between positively charged cationic liposomes and negatively charged delivered mRNA molecules (N/P ratio) is an important parameter that affects the efficiency of NA delivery [8,24]. An optimal N/P ratio should lead to the complete binding of RNA into lipoplexes and the neutralization of negatively charged RNA, resulting in its protection from nuclease degradation. A net-positive surface charge must be, on the one hand, sufficient for lipoplexes to effectively interact with the cell membrane and for internalization, and, on the other hand, not too high to prevent cellular toxicity and aggregation when interacting with negatively charged serum proteins (albumins) [25,26].

In our work, to create an effective liposomal mRNA delivery system for eukaryotic cells, we studied the effect of cationic liposome composition based on cationic amphiphile 2X3 and zwitterionic lipid DOPE, as well as the N/P ratio of the physicochemical characteristics and morphology of lipoplexes formed. Moreover, we considered the efficiency of mRNA delivery into eukaryotic cells.

## 2. Materials and Methods

### 2.1. Preparation of Cationic Liposomes

For the experiments, 1,26-bis(cholest-5-en-3β-yloxycarbonylamino)-7,11,16,20-tetraazahexacosane tetrahydrochloride (2X3) (Figure 1) [9] and 1,2-dioleoyl-sn-glycero-3-phosphoethanolamine (DOPE) were used. The characterization of 2X3 was performed using NMR (Appendix A). DOPE was purchased from Avanti Polar Lipids (Alabaster, AL, USA). A solution of 2X3 in a mixture of CHCl–CH_3_OH (1:1 vol.) was added to a solution of DOPE in CHCl_3_ at molar ratios of 1:1, 1:2, and 1:3 and gently stirred.

Organic solvents were removed in vacuo, and the lipid film obtained was dried for 4 h at 0.1 Torr to remove residual organic solvent. Then, it was hydrated using deionized water (MilliQ, Burlington, MA, USA) at 4 °C overnight. The liposomal dispersion was sonicated for 15 min at 70–75 °C in a bath-type sonicator (Bandelin Sonorex Digitec DT 52H, Berlin, Germany), filtered (0.45 μm Chromafil^®^ CA-45/25; Macherey–Nagel, Düren, Germany), flushed with argon, and stored at 4 °C.

### 2.2. In Vitro Transcription

mRNAs encoding eGFP (mRNA-eGFP) and firefly luciferase (mRNA-FLuc) were obtained by in vitro transcription using the mMESSAGE mMACHINE kit (Invitrogen, #AM1344, Waltham, MA, USA). The pJET1.2-eGFP (Thermo Scientific, #K1231, Waltham, MA, USA) plasmid, containing the T7 promoter and the eGFP gene, and the pMax-Luc_FF plasmid, containing the T7 promoter and the luc-ff gene, were used as templates for RNA synthesis during in vitro transcription. One microgram of linearized plasmid template was used for the reaction. Anti-Reverse Cap Analog (ARCA) (Thermo Fisher Scientific, #AM8045) was used for efficient translation of the RNA. To maximize RNA yield and the fraction of capped transcripts, we used a cap ARCA/GTP ratio of 4:1. The reaction was performed according to the manufacturer’s protocol. Following RNA synthesis, the DNA template was removed via subsequent digestion with DNase Turbo (Invitrogen, #AM1345). The Poly(A) Tailing Kit (Invitrogen, #AM1350) was used to polyadenylate the 3′-termini of the transcribed RNA. The transcript was purified via lithium chloride precipitation according to the recommended protocol (Invitrogen, #AM1345). The RNA sample was analyzed using electrophoretic separation under denaturing conditions (Appendix A). The RNA sample (500 ng) was mixed with an equal volume of Gel Loading Buffer II (Invitrogen, #AM1345) and heated for 5 min at 80 °C. Samples were subsequently loaded into wells of 1% agarose gel (containing 0.5 µg/mL ethidium bromide) and run in 1× MOPS buffer at room temperature.

### 2.3. Preparation of Complexes of Cationic Liposomes with mRNA

To measure the physicochemical characteristics of the complexes, 125 ng of mRNA in 2.5 μL distilled water was mixed with 2.5 μL of cationic liposomes in distilled water at N/P ratios of 1/1, 2/1, 4/1, 6/1, 8/1, 10/1, 20/1, and 30/1. Then, it was incubated for 20 min at 25 °C and diluted with the required volume of distilled water.

For transfection experiments, complexes of cationic liposomes with mRNA were formed in MEM medium (Gibco, Waltham, MA, USA) without serum by mixing 5 µL of an mRNA solution (0.1 μg) and 5 μL of 1 mM liposome solution in DMEM at N/P ratios of 4/1, 6/1, 8/1, 10/1, and 20/1, with vortex mixing for 10 s followed by incubation for 20 min at 25 °C.

### 2.4. Size and Zeta Potential Measurement

Solutions of liposomes and lipoplexes were prepared as described above in distilled water. Particle size was determined via dynamic light scattering, and zeta potentials were measured with the electrophoretic method using the Zetasizer 3000 HS (Malvern Instruments, Malvern, UK) (angle—173°; viscosity—0.890 cP; temperature—25 °C; equilibrium time—2 min; refractive indices—1.33 and 1.45 for water and liposomes, respectively). Results are expressed as the average of three measurements.

### 2.5. Transmission Electron Microscopy (TEM)

Samples were prepared according to a negative staining protocol. Liposome solution in water (20 μL) was applied to a formvar-coated 300-mesh microscopy grid (Electron Microscopy Science, Hatfield, PA, USA) for 30 s, followed by washing in water (twice for 30 s). Samples were contrasted for 30 s with 1.5% aqueous solution of phosphotungstic acid sodium salt. Measurements were performed using a JEM 1100 electron microscope (JEOL, Tokyo, Japan) with an accelerating voltage of 80 kV.

### 2.6. Atomic Force Microscopy (AFM)

Solutions of liposomes were prepared as described above in nuclease-free water. Water (30 μL) was added to 1 µL of a liposome solution, or to 5 µL liposome/mRNA complexes, applied to a freshly cleaved mica (SPI Supplies, West Chester, PA, USA), and incubated at a room temperature for 1 min. The solution was discarded and the mica surface was washed with water and dried with a Concentrator Plus (Eppendorf, Hamburg, Germany). Sample surface topography was measured in semicontact mode on the SolverNext scanning probe microscope (NT-MDT, Moscow, Russia) using an NSG03 probe (NT-MDT, Moscow, Russia). Images were processed using Gwyddion 2.62 software (Czech Metrology Institute, Brno, Czech Republic) [27], and particle height (Z, nm) and equivalent disk radius (R_eq_, nm) were calculated.

### 2.7. Gel Retardation Assay with Capillary Electrophoresis

The mRNA binding activity of 2X3-DOPE cationic liposomes was determined by capillary electrophoresis using the Agilent RNA 6000 Nano Kit (#5067-1511) on an Agilent 2100 Bioanalyzer (Agilent Technologies, St. Clara, CA, USA). The chip was primed and loaded as described by the manufacturer. After electrophoresis, the results were analyzed with the 2100 Expert software version B.02.08.SI648 (SR2) (Agilent Technologies, St. Clara, CA, USA).

### 2.8. Quantification of Encapsulation Efficiency

The mRNA loading in liposomal formulations was quantified using a Quant-iT RiboGreen assay (Thermo Fisher, Oxford, UK). Samples and standard solutions were prepared in water. The assay was performed according to the manufacturer’s protocol. Samples were loaded on a black 96-well plate and analyzed for fluorescence on a microplate reader (ClarioStar, BMG Labtech, Ortenberg, Germany) at an excitation of 485 nm and emission of 528 nm.

### 2.9. Cell Lines

BHK-21 cell were purchased from ATCC, CCL10, and cultured at 37 °C in a humidified incubator with 5% CO2 in MEM medium (Gibco, USA) supplemented with 4 mM L-glutamine, 1 mM sodium pyruvate, and 10% fetal bovine serum (FBS) (Biowest, Nuaillé, France). A549 cells were purchased from ATCC, CCL-185, and cultured at 37 °C in a humidified incubator with 5% CO2 in DMEM medium (Biolot, St. Petersburg, Russia) and 10% fetal bovine serum (FBS).

### 2.10. Cell Transfection

The day before transfection, BHK-21 and A549 cells were cultured in 96-well plates under normal growth conditions (density 15 × 103 cells per well). On the day of transfection, complexes of cationic liposomes with mRNA in a volume of 10 μL were added to the cells and were incubated until analysis.

### 2.11. Fluorescence Microscopy

Plates were imaged with a 10× objective using the Cytell™ Cell Imaging System (GE Healthcare Life Sciences, Little Chalfont, UK). Nuclei were stained with NucBlue (Thermo Fisher Scientific, USA). Images were processed using ImageJ version 1.53u (National Institutes of Health, Bethesda, MD, USA) (https://imagej.nih.gov/ij/index.html, accessed on 1 November 2022).

### 2.12. Flow Cytometry Analysis

The transfection of mRNA-eGFP was performed as described above in 96-well plates. After the incubation of cells for 24 h with lipoplexes, cells were removed with 0.05% trypsin-EDTA (Gibco, USA) and washed with PBS solution. To detect the level of GFP expression, 10,000 events per sample were processed in separate cells and obtained using a CytoFLEX flow cytometer (Beckman Coulter, Indianapolis, IN, USA). Transfection efficiency was quantified using flow cytometry based on two parameters: percent transfection (%) and mean fluorescence intensity (MFI). Percent transfection was calculated as the percentage of GFP-positive cells. The MFI was calculated as a median for gated GFP-positive cells. The results were analyzed in Kaluza software and were expressed as the mean and standard deviation obtained from three samples.

### 2.13. Luminescence Analysis

After 24 h from the initial transfection of FLuc-coding mRNA, 50 μL of medium was removed from each well and 50 μL of Dual-Glo^®^ Luciferase Assay System substrate (Promega, Southampton, UK) was added. It was incubated for 10 min at 25 °C and analyzed on a microplate reader (ClarioStar, BMG Labtech, Germany).

### 2.14. Cytotoxicity Studies

MTS [3-(4,5-dimethylthiazol-2-yl)-5-(3-carboxymethoxyphenyl)-2-(4-sulfophenyl)-2H-tetrazolium, inner salt] (Promega, UK) assays were performed with cationic liposomes in concentrations equivalent to those used for mRNA delivery. Briefly, 20 μL MTS was added to the samples and incubated for 2 h. The absorbance was measured using a plate reader at 490 nM. The percentage of cell viability of the test sample was calculated with the following formula: A test/A control × 100.

## 3. Results and Discussion

### 3.1. Preparation and Characterization of Liposomes: Effect of Cationic Lipid Composition on Lipoplex Formation with mRNA

Previous studies have shown that cationic liposomes 2X3-DOPE can effectively deliver nucleic acids in vitro and in vivo [10,11,12,13,28]. It has previously been demonstrated that the molar ratios of polycationic amphiphile 2X3 and helper lipid DOPE in the structure of liposomes affect the efficiency of pDNA transfection [13]. The use of DOPE in liposomes leads to the efficient delivery of nucleic acids due to its high fusogenic nature and the ability to form an inverted hexagonal phase under conditions of endosome acidification [29]. Many early studies have shown that DOPE promotes a more compact packaging of nucleic acids [19,30].

For our study, we prepared cationic liposomes based on 2X3 and DOPE with an increase in molar ratio towards the helper lipid: 2X3-DOPE 1:1, 2X3-DOPE 1:2, and 2X3-DOPE 1:3. Cationic liposomes were obtained by the dry film method, followed by sonication according to the previously described method [12,13,14].

Physicochemical characteristics (particle size and ζ potential) are considered the key parameters in determining the efficiency of cationic liposomes as delivery vehicles for nucleic acids [31,32]. The hydrodynamic diameters of the liposomes and ζ potential were characterized by dynamic light scattering (Table 1). All liposomes had a size <100 nm, PDI below 0.3, and a surface charge above +40 mV.

According to the obtained results, the hydrodynamic diameter did not correlate with the change in the molar ratios of 2X3 and DOPE, and was <100 nm. The ζ potential of liposomes 2X3-DOPE 1:2 and 2X3-DOPE 1:3 was greater than for 2X3-DOPE 1:1. An increase in the amount of neutral lipid DOPE to cationic lipid 2X3 led to an increase in ζ potential. DOPE is a lipid that has a “conical shape” and does not form a lamellar phase; as a result, DOPE forms phases with a large radius of curvature [22]. Consequently, an increased amount of DOPE in the lipid mixture may lead to its greater distribution in the inner layer of vesicles, while the cationic lipid 2X3 will prevail in the surface layer, which, in turn, will increase the surface charge of the liposome.

Another important variable which effects transfection efficiency is the physicochemical properties of the cationic lipid [13,33]. Liposome morphologies were characterized using TEM (Figure 2). We observed two types of lipid particles in the 2X3-DOPE 1:1 sample: spherical particles (d ~100 nm) and rod-like particles (approximately 10 nm wide and 20 to 500 nm long). In the 2X3-DOPE 1:2 sample, mostly spherical particles were observed, the diameter of which varied from 100 to 500 nm. In the 2X3-DOPE 1:3 sample, as well as in 2X3-DOPE 1:2, there were no rod-like particles, and only clumped liposomes, close to a spherical shape (d ~50–55 nm), were observed. The TEM results are consistent with the results of DLS and indicate the predominance of liposomes with a diameter of approximately 100 nm in the samples. It should be noted that, according to TEM data, rod-like particles were observed only at the lowest DOPE helper lipid content (2X3-DOPE 1:1) and were absent with an increase in concentration (in samples 2X3-DOPE 1:2 and 2X3-DOPE 1:3). Thus, the number of rod-like particles decreased with a decrease in the molar fraction of the cationic lipid 2X3. Cationic amphiphilic lipid 2X3 contains two hydrophobic cholesterol residues [10]. It has previously been shown that these structures can induce the formation of inverted non-lamellar phases in lipid systems [22,23]. It can be assumed that a decrease in the molar ratio of cationic lipid 2X3 in the composition of liposomes 2X3-DOPE leads to a decrease in the number of inverted structures. This assumption is confirmed by the work of Angelov et al., who showed that the high molar content of double-chain cationic lipid 1,2-dimyristoly-sn-glycero-3-ethylphosphocholine (EPC14) leads to the formation of inverted hexagonal structures (hexosomes) and a resultant analysis structure of MO/EPC14/DOPE-PEG2000 via the SAXS method (MO, neutral monoglyceride lipid monoolein) [33].

Next, we studied the physicochemical characteristics of the liposomes/mRNA complexes formed at various N/P ratios (Figure 3). The obtained data show that mRNA encapsulation via 2X3-DOPE cationic liposomes depends on both the amount of DOPE and the nature of the studied RNA molecules. Thus, 2X3-DOPE 1:1 and 2X3-DOPE 1:2 formed smaller complexes with mRNA-eGFP than with mRNA-Fluc at almost all N/P ratios. In contrast, lipoplexes based on 2X3-DOPE 1:3 and mRNA-FLuc were smaller than with mRNA-eGFP for almost all N/P ratios. The smallest particles (87 nm) were observed in complex 2X3-DOPE 1:3 with mRNA-FLuc at a N/P ratio of 10/1, while the lipoplexes with mRNA-eGFP had a size of 142 nm at the same N/P ratio. Interestingly, when comparing the size of 2X3-DOPE 1:3 lipoplexes with mRNAs at the same N/P ratio of 10/1, we observed a significant difference. It is known that mRNA is a single-strand molecule with exposed nucleobases, which makes it significantly hydrophobic. This leads to additional hydrophobic interactions with liposome lipids [8]. Based on our results, a more compact complex formation can be observed by increasing the proportion of DOPE and mRNA length. At the same time, the ζ potential of the lipoplexes did not differ much between the complexes with mRNA-eGFP and mRNA-FLuc for the corresponding ratios of lipids and N/P. For all lipoplexes with an N/P ratio of 4/1 or more, the values of the ζ potentials were above +30 mV.

Lipoplexes were analyzed by the capillary electrophoresis method to determine the N/P ratio at which the cationic liposomes completely bound to the mRNA-eGFP molecules (Figure 4a). It was shown that, starting from an N/P ratio of 4/1, the band corresponding to mRNA disappeared, indicating the complete binding of mRNA with liposomes.

The mRNA-eGFP loading efficiency of lipoplexes was also studied using an indirect method of free mRNA concentration determination after lipoplex formation using ribogreen (Figure 4b). The ribogreen assay results are consistent with the results of the capillary electrophoresis: RNA is completely encapsulated by liposomes from the N/P ratio of 4/1.

AFM was performed to visualize changes in the morphology of liposomes upon binding to mRNAs. For the analysis, an N/P ratio of 10/1 was chosen as the optimal ratio at which both negatively charged nucleic acid molecules were completely bound and the positive charge was retained, which is necessary for effective cell penetration. Such a ratio demonstrated maximum efficiency in experiments in vitro (see Section 3.2).

Figure 5a shows representative images of the liposome samples’ surface topography (‘2X3-DOPE 1:1’, ‘2X3-DOPE 1:2’, and ‘2X3-DOPE 1:3’) and liposome–mRNA complexes (‘lipoplexes’). To quantify the observed changes in the morphology of the liposomes, the heights of the liposomes and the radii of the equivalent disks were calculated based on the AFM images. As a result, the diameters of the equivalent spheres (D_afm_, nm) were obtained (Figure 5b, Table 2). It is seen that, with an increase in DOPE content, the liposome size D_afm_ decreased from 200 to 100 nm (Figure 5b, Table 2), and polydispersity was retained. When forming mRNA–liposome complexes, particle size significantly decreased to a similar diameter ~50–60 nm for all 2X3-DOPE ratios. Thus, only one characteristic image is presented for all the lipoplexes studied. It should be noted that there were no significant differences between sizes of the complexes with mRNA-eGFP and mRNA-FLuc when using 2X3-DOPE 1:3.

The particle size decreases during complexation, which may indicate the formation of compact complexes containing mRNA. Interestingly, the resulting particle size after complexation depends on the 2X3-DOPE ratio: an increase in DOPE concentration leads to an increase in lipoplexes size. We observed the opposite in the case of pure liposomes. It follows that the size changes in 2X3-DOPE 1:1 liposomes during the formation of complexes with both mRNAs were the largest (from 200 to 40–50 nm), while those of 2X3-DOPE 1:3 were the smallest (from 100 to 60 nm). According to the obtained AFM results, the encapsulation of both mRNA-eGFP and mRNA-FLuc led to all the lipoplexes acquired being of a similar size, despite the significant difference in the initial sizes of liposomes before binding (Table 2, Figure 5). Thus, it can be assumed that 2X3-DOPE 1:3 underwent the smallest structural alterations when binding RNA, since their changes in size were also the smallest. Based on these data, we can hypothesize that, when binding with 2X3-DOPE 1:3, mRNA is subjected to the least impact that can provoke greater transfection efficiency.

### 3.2. Effect of Cationic Lipid Composition on mRNA Delivery into Eukaryotic Cells

The characterized lipoplexes were tested for model mRNA-eGFP delivery efficiency into the BHK-21 cell line (Figure 6). mRNA-eGFP was chosen as a quantitative model to evaluate transfection efficiency based on two parameters: percent transfection (%), and mean fluorescence intensity (MFI). Percent transfection corresponds to the percentage of cells that have been successfully transfected, while MFI is a measure of the intensity of GFP expression per cell, and corresponds to the extent to which each cell has been transfected and the efficacy of mRNA release into the cytoplasm.

In the previous stage, we showed that, starting from an N/P ratio of 4/1, all the lipoplexes ranged in size up to 250 nm, had positive ζ potential (Figure 3), and completely encapsulated mRNA (Figure 4). Based on the data obtained, we have chosen the N/P ratios of 4/1, 6/1, 8/1, 10/1, and 20/1 to measure the transfection efficiency of mRNA. The cationic liposomes 2X3-DOPE 1:1 and 2X3-DOPE 1:2 demonstrated the highest transfection efficiency (both in terms of % transfection and MFI) of mRNA-eGFP at an N/P ratio of 10/1 (Figure 6b). At the same time, 2X3-DOPE 1:3 demonstrated similar efficiency in terms of % transfection at N/P ratios from 4/1 to 10/1; furthermore, at a ratio of 20/1, the average fluorescence intensity was significantly higher than when using the commercial transfection reagent Lipofectamine MessengerMAX. It can be concluded that an increase in the helper lipid DOPE in cationic liposome composition leads to an increase in the transfection efficiency of mRNA. The lipoplexes with 2X3-DOPE 1:3 demonstrated the highest transfection efficiency in terms of the mean fluorescence intensity (Figure 6a,b).

We also examined the transfection efficiency of mRNA encoding firefly luciferase (FLuc) delivered by lipoplexes (see Figure 6c). All lipid compositions possessed the highest transfection efficiency at an N/P ratio of 10/1; moreover, they were significantly more effective than the commercial transfection reagent Lipofectamine MessengerMAX. The other N/P ratios had a lower transfection efficiency. At the same time, 2X3-DOPE 1:3 demonstrated maximum transfection efficiency at an N/P ratio of 10/1.

We evaluated the transfection efficiency of the studied liposomes at an N/P ratio of 10/1 in A549 cells (Figure 7), since this ratio has been shown to be the most effective (Figure 6). The percentage of transfected cells in the case of 2X3-DOPE 1:3 was 8.8 ± 0.1%, compared with Lipofectamine MessengerMAX, which was 42.7 ± 1.5%, but the values of the mean fluorescence intensity were comparable (1.1 ± 0.3 × 10^5^ rfu and 1.4 ± 0.2 ×10^5^ rfu, respectively). The transfection efficiency of FLuc-coding mRNA delivered by lipoplexes based on 2X3-DOPE 1:3 was much higher than the commercial transfection reagent, and this was also demonstrated in BHK-21 cells.

In our previous study, we showed that cationic liposomes based on 2X3 and DOPE were non-toxic to eukaryotic cells [34]. The studied cationic liposomes are non-toxic to BHK-21 and A549 cells in concentrations used for the most effective N/P ratio (10/1) for mRNA delivery (Appendix A).

The lipid molar ratio 2X3-DOPE is among the key parameters determining the transfection efficiency of NA. We showed that transfection efficiency correlated with the data obtained via DLS: lipoplexes demonstrate the highest efficiency with an increase in DOPE and mRNA length, and 2X3-DOPE 1:3 lipid composition provides both effective transfection and protein expression at an N/P ratio of 10/1 for BHK-21 and A549 (Figure 6 and Figure 7). It has previously been shown that 2X3-DOPE liposomes with a molar ratio of 2:1 demonstrate the best efficiency in pDNA delivery [13]. The differences in the molar ratios of 2X3-DOPE for the efficient intracellular delivery of mRNA and pDNA may be because mRNA molecules are more amphiphilic than pDNA molecules [8]. Interestingly, the opposite result was obtained by comparing the transfection of pDNA and mRNA by liposomes based on the cationic lipid aminolipophosphonate and DOPE [21]. They showed that maximum mRNA transfection efficiency occurs with a decrease in the molar ratio of DOPE and an increase in the molar ratio of cationic lipids, while the maximum efficiency of pDNA transfection, on the contrary, is observed with a decrease in the molar ratio of cationic lipids.

We also found that the transfection efficiency of mRNA-eGFP using 2X3-DOPE 1:3 was comparable to the commercial reagent (in terms of MFI), while that of mRNA-FLuc was much higher. The difference in transfection efficiency between 2X3–DOPE 1:3 mRNA-eGFP and 2X3–DOPE mRNA-FLuc at a N/P ratio of 10/1 can be partially related to the smaller size of the latter (Figure 3, Figure 6 and Figure 7). It has previously been shown that mRNA transfection efficiency and mRNA translation can be determined by the ratio of cationic lipid and DOPE in the liposome. Akhter et al. showed that when mRNA interacts with cationic liposomes, irreversible conformational changes occur in its secondary structure, leading to a decrease in translational activity [21]. The molecular weight of mRNA-eGFP (960 nt + ~100 nt poly(A) tail) and mRNA-FLuc (1679 nt + ~100 nt poly(A) tail) and their secondary structures are different. It can be hypothesized that cationic liposomes have different effects on RNA secondary structure during complex formation. The longer the delivered mRNA molecules, the greater the molar ratio of DOPE in liposome compositions, the smaller the conformational change, and the higher the transfection efficiency of mRNA.

It is worth noting that when mRNA-eGFP was transferred to BHK-21 cells, the transfection efficiency (% transfected cells and MFI) of 2X3-DOPE 1:3 showed no significant change at N/P ratios from 4/1 to 10/1 (Figure 6). The most effective ratio in terms of MFI was a N/P ratio of 20/1. At the same time, delivery of the longer mRNA-FLuc with increasing N/P ratios from 4/1 to 10/1 resulted in an increase in transfection efficiency, and the maximum transfection efficiency was demonstrated at an N/P ratio of 10/1. This once again confirms the role of mRNA length and structure in transfection efficiency when using 2X3-DOPE liposomes.

Our results, in line with previous studies, show that 2X3-DOPE ratios determine the structure and physicochemical properties of liposomes, upon which the encapsulation and transfection of various nucleic acids depend.

## 4. Conclusions

In this study, we demonstrated the effect of several variables on the efficiency of mRNA delivery, using polycationic lipid 2X3 and helper lipid DOPE as a delivery system. The molar ratio of cationic lipids and helper lipids in the liposome composition, the N/P ratio, the structure of mRNA, and the potential ability of DOPE to form inverted structures in endosomes are the most important parameters that should be considered when optimizing and developing an effective delivery system based on cationic lipid 2X3. An increase in the proportion of DOPE in liposome formulations, based on 2X3, increases the efficiency of mRNA delivery to eukaryotic cells, which can be explained by the interactions between the cationic lipid and mRNA molecules, and this affects the secondary structure of the latter. Liposomes based on the cationic lipid 2X3 and neutral helper lipid DOPE can be offered as effective delivery vehicles for the future development of mRNA vaccines.

## Figures and Tables

**Figure 1 pharmaceutics-15-00008-f001:**
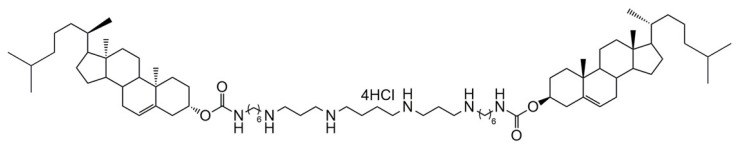
Structural formula of the cationic lipid 2X3.

**Figure 2 pharmaceutics-15-00008-f002:**
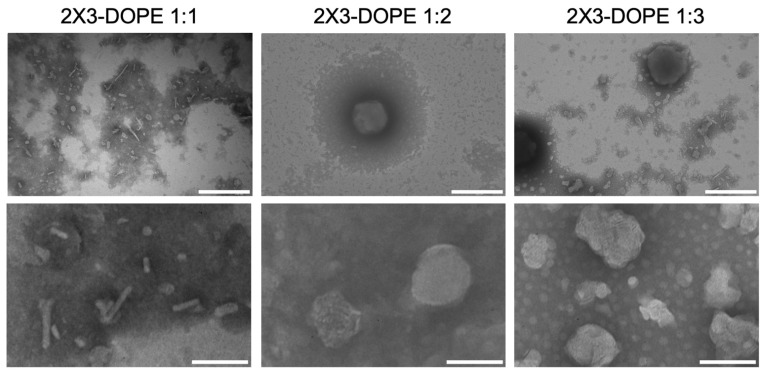
TEM images of 2X3-DOPE 1:1, 2X3-DOPE 1:2, and 2X3-DOPE 1:3 liposomes in water. Scale bar is 500 nm (upper panel) or 100 nm (lower panel).

**Figure 3 pharmaceutics-15-00008-f003:**
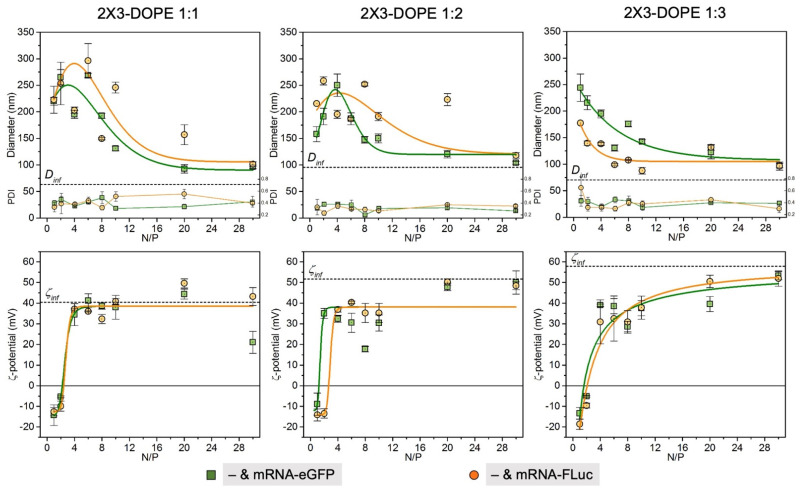
Physicochemical characteristics of lipoplexes with mRNA-eGFP and mRNA-FLuc in different N/P ratios.

**Figure 4 pharmaceutics-15-00008-f004:**
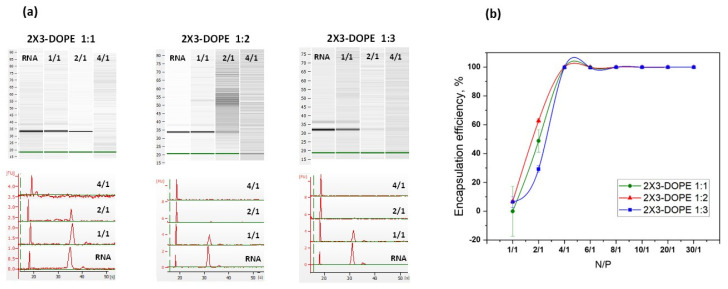
Liposomes and mRNA complex formation. (**a**) Capillary electrophoresis method. Electropherograms of free mRNA (RNA) and lipoplexes at N/P ratios of 1/1, 2/1, and 4/1. (**b**) Ribogreen assay for quantification of encapsulated mRNA.

**Figure 5 pharmaceutics-15-00008-f005:**
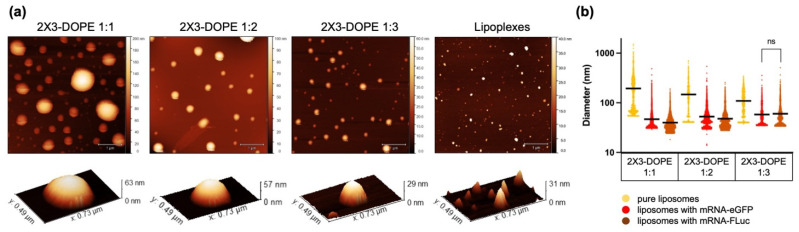
(**a**) Surface topography of 2X3-DOPE 1:1, 2X3-DOPE 1:2, and 2X3-DOPE 1:3 liposomes without RNA, and the characteristic image of 2X3-DOPE complexes with mRNA (both eGFP and FLuc) at an N/P ratio of 10/1 (‘lipoplexes’). The scale bar is 1 µm. Pseudo-color rulers, reflecting the particle height distribution in each sample, are located on the right side of each image. The bottom row shows the characteristic particles of each sample in 3D view. (**b**) Diameter of the particles calculated based on AFM data. ‘ns’—not significant difference (*p*-value > 0.05 based on one-way ANOVA statistics with correction for multiple comparisons using Tukey’s test); in all other cases, the samples significantly differ (*p* < 0.0001).

**Figure 6 pharmaceutics-15-00008-f006:**
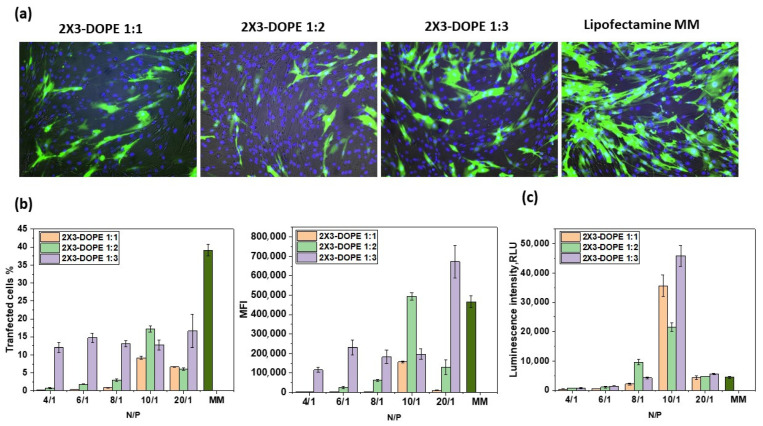
Transfection efficiency of mRNA using cationic liposomes 2X3-DOPE 1:1, 2X3-DOPE 1:2, and 2X3-DOPE 1:3 in BHK-21 cells 24 h after transfection. (**a**) Fluorescent microscopy results for mRNA-eGFP transfection at an N/P ratio of 10/1; green—eGFP protein; blue—cell nuclei. (**b**) Flow cytometry analysis of eGFP expression in transfected cells: the percentage of eGFP-positive cells (“transfected cells, %”, left) and the mean fluorescence intensity (MFI) of the cells (right) vs. the N/P ratio. (**c**) Transfection efficiency of FLuc-coding mRNA delivered by lipoplexes in different N/P ratios. Luciferase expression is expressed as relative light units (RLUs). MM is a positive control for mRNA transfection (Lipofectamine MessengerMAX reagent).

**Figure 7 pharmaceutics-15-00008-f007:**
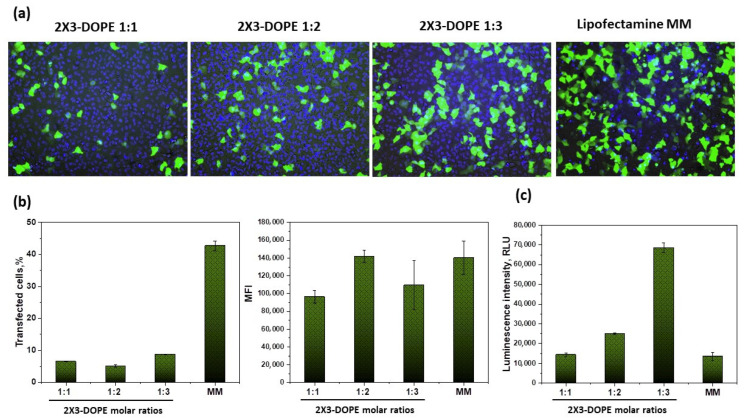
Transfection efficiency of mRNA-eGFP using cationic liposomes 2X3-DOPE 1:1, 2X3-DOPE 1:2, and 2X3-DOPE 1:3 at an N/P ratio of 10/1 in A549 cells 24 h after transfection. (**a**) Fluorescent microscopy results; green—eGFP protein; blue—cell nuclei. (**b**) Flow cytometry analysis of eGFP expression in transfected cells: the percentage of eGFP-positive cells (“transfected cells, %”, left) and the mean fluorescence intensity (MFI) of the cells (right). (**c**) Transfection efficiency of FLuc-coding mRNA delivered by lipoplexes. Luciferase expression is expressed as relative light units (RLUs). Molar ratios of 1:1, 1:2, and 1:3 of 2X3 and DOPE, respectively. Lipofectamine MessengerMAX reagent (MM) is a positive control for mRNA transfection.

**Table 1 pharmaceutics-15-00008-t001:** Composition of cationic liposomes, and their physicochemical parameters expressed as mean ± SD.

Sample	Size (nm)	PDI	ζ Potential (mV)
2X3-DOPE 1:1	62.7 ± 0.1	0.23 ± 0.01	40.4 ±1.2
2X3-DOPE 1:2	95.6 ± 0.7	0.23 ± 0.01	51.7 ± 8.0
2X3-DOPE 1:3	70.5 ± 0.4	0.26 ± 0.00	58.1 ± 2.0

**Table 2 pharmaceutics-15-00008-t002:** Particle size D_afm_ (nm) in the samples of pure liposomes (“pure”) and liposome–mRNA complexes (‘& mRNA-eGFP’ and ‘& mRNA-FLuc’), calculated from AFM images (scanning areas 20 x 20 μm). Data are presented as the mean ± SD. D_afm_—the diameter of the equivalent sphere, calculated based on the equivalent disk radius and height.

	2X3-DOPE 1:1	2X3-DOPE 1:2	2X3-DOPE 1:3
Pure	195 ± 183	147 ± 116	109 ± 62
& mRNA-eGFP	47 ± 24	53 ± 27	58 ± 30
& mRNA-FLuc	40 ± 17	48 ± 24	60 ± 35

## Data Availability

The data that support the findings of this study are available from the corresponding author, V.V., upon reasonable request.

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
