# Peer review of "Influence of Lipid Composition of Cationic Liposomes 2X3-DOPE on mRNA Delivery into Eukaryotic Cells"

_pharmaceutics, 2022, doi:10.3390/pharmaceutics15010008_

Round 1
Reviewer 1 Report
The paper entitled Influence of lipid composition of cationic liposomes 2X3-DOPE 2 on mRNA delivery into eukaryotic cells is well written. The authors explored the influence of the helper lipid DOPE and the N:P ratio on liposome formation, and transfection efficiency. The authors also compared two different mRNA and studied the effect of mRNA length on particle performance.
Despite the rigorous work, novelty remains a big issue. The reviewer believes that the paper lacks novelty, and is best suited to another journal. Alternatively, the authors are invited to complement their studies with in vivo experiments to demonstrate that this liposomal formulation is efficient and nontoxic or at least address the below points before re-consideration into Pharmaceutics.
Few suggestions:
1- The structure of the 2X3 cationic lipid needs to be added.
2- The characterization data (e.g., NMR, MS) of the 2X3 cationic lipid and DOPE needs to be added to supplemental information.
3- Characterization of the mRNA should be performed as the traces on the bioanalyzer shows multiple bands. The quality of the mRNA can affect the liposome data.
4- The author must clarify why the surface charge of the particle is increasing with increased molar ratio of DOPE; the latter should be neutral at the pH where the measurement has been done.
5- The author must explain how the mRNA is becoming encapsulated into the liposomes following a simple mixing in water. The data from the AFM shows that the 2X3-DOPE forms spherical particles (or empty liposomes). In their method, the author mix the 2X3-DOPE complexes with mRNA, and wait for 20 minutes. Therefore, the mechanism by which the mRNA is becoming encapsulated needs to be either elucidated or explained.
Thank you
Reviewer 2 Report
The review manuscript by Vera V. et al., entitled “Influence of lipid composition of cationic liposomes 2X3-DOPE 2 on mRNA delivery into eukaryotic cells” developed liposomal formulations with their previously reported ionizable lipid 2X3 and varying molar ratios of DOPE as helper lipid in order to signify the influence of components in the liposomal formulations on their transfection properties. However, there are multiple studies in this lines, probably, the behavior of 2XL3 in varying concentrations of DOPE is worth for an investigation. There are several concerns, listed below needs to be addressed. Overall, this manuscript is not recommended for publication in its current form and may be considered for publication after the revision.
Major Comments:
1. Surprisingly, eGFP mRNA transfections with the liposomal formulations showed below 20% in both BHK21 and A549 cells, whereas, with messenger Max showed above 50% transfection efficiency. The phenomenon was completely contrast with Luciferase mRNA transfections. I agree that size of mRNA may affect transfection efficiency but not this dramatic level. Authors should properly explain why such low levels of eGFP mRNA transfections were observed with the liposomes despite having the similar particles sizes and surface potentials with both the mRNAs.
2. Luminescence assay is usually more sensitive than eGFP expression. Contrastingly, in figures 5 & 6, eGFP expression found to be more sensitive than luciferase expression.
3. In figure 5a, transfection efficiencies of 2X3 DOPE 1:3found to be at least 2 fold lower compared to messenger max, but the MFI found to be around 1.5 fold higher. Authors should explain how they calculated MFI?
4. Authors should perform cytotoxicity studies to demonstrate that the liposomes are safe for NA transfections.
Minor Comments:
1. BHK21 cells should be defined and abbreviation should be consistent throughout the manuscript.
2. Authors mentioned only BHK21 cells in the abstract but not A549 cells. They should mention in the abstract as well.
3. Composition of the liposomal formulations for efficient transfections vary with the cell types as their endocytosis process differ. Hence, Authors should perform transfection experiments at least in 2 more different cell types to confirm the effectiveness of the developed formulation.
Reviewer 3 Report
Influence of lipid composition of cationic liposomes 2X3-DOPE on mRNA delivery into eukaryotic cells.
Manuscript ID: 2074650
The authors have designed lipoplexes based on polycationic lipid 1,26-bis(cholest-5-en-3β-yloxycarbonylamino)-7,11,16,20-tetraazahexacosane tetrahydrochloride (2X3) and helper lipid, 1,2-dioleoyl-sn-glycero-3-phosphoethanolamine (DOPE) in different molar ratios (1:1, 1:2, 1:3) were formulated to efficiently deliver of model mRNAs to BHK cells. The authors have examined the effect of the 2X3-DOPE composition as well as lipid to mRNA ratio.
The study is well designed and presented in the present manuscript. The introduction, discussions, conclusion and results sections have been elaborated correctly. The manuscript can be accepted in the current form.
Author Response
We appreciate for your review.
Round 2
Reviewer 1 Report
The authgors responded with reason to the questions. More experiments is warranted. However, i will consider myself satisfied
Reviewer 2 Report
The Authors have addressed most of my previous comments. However, The introductions still needs to be improved further with grammar check.
The manuscript is recommended for publication after the suggested revision.